# Macrocyclic Receptors for Identification and Selective Binding of Substrates of Different Nature

**DOI:** 10.3390/molecules26175292

**Published:** 2021-08-31

**Authors:** Galina Mamardashvili, Nugzar Mamardashvili, Oscar Koifman

**Affiliations:** G.A. Krestov Institute of Solution Chemistry of the Russian Academy of Sciences, Akademicheskayast. 1, 153045 Ivanovo, Russia; gmm@isc-ras.ru (G.M.); president@isuct.ru (O.K.)

**Keywords:** porphyrin, receptor, molecular recognition, host–guest interactions, binding cavities

## Abstract

Molecular recognition of host/guest molecules represents the basis of many biological processes and phenomena. Enzymatic catalysis and inhibition, immunological response, reproduction of genetic information, biological regulatory functions, the effects of drugs, and ion transfer—all these processes include the stage of structure recognition during complexation. The goal of this review is to solicit and publish the latest advances in the design and sensing and binding abilities of porphyrin-based heterotopic receptors with well-defined geometries, the recognition ability of which is realized due to ionic, *H*-bridge, charge transfer, hydrophobic, and hydrophilic interactions. The dissection of the considered low-energy processes at the molecular scale expands our capabilities in the development of effective systems for controlled recognition, selective delivery, and prolonged release of substrates of different natures (including drugs) to their sites of functioning.

## 1. Introduction

Molecular recognition is a process in which host molecules (receptors) select and bind guest molecules (substrates) into structurally highly organized systems through multipoint intermolecular interactions [1,2,3,4,5,6,7]. Porphyrins are extremely useful for this purpose. Their unique properties are the result of the unusual geometric and electronic structure of the porphyrin macrocycle, containing a developed aromatic conjugated system. Selective chemical modification of porphyrins by fragments of other classes of compounds makes it possible to synthesize molecular systems that are different in their nature and positions of the reaction centers relative to each other [8,9,10,11,12,13,14]. This review discusses the progress made so far in the design and sensing and binding abilities of porphyrins modified with calix[4]arenes, calix[4]pyrroles, oxacalix-[2]-arenes-[2], and resorcinarenes and with bulky carbazolylphenyl substituents of different generations towards substrates of different natures. The molecular structures, sensing mechanism, and successful applications of these macrocyclic compounds with well-defined geometries as heterotopic receptors for molecular recognition and reversible binding of anions, cations, and small organic molecules are discussed in detail. Data on the ion-assisted binding ability of tetrapyrrole macroheterocyclic compounds decorated with additional complexing cavities and channels are of particular interest. It is shown that the binding ability of the porphyrin core depends on the conformation mobility of the substituents and the possibility of formation of intramolecular cavities and channels of various shapes within them. We hope this review will bring more attention to this rising field and inform researchers in related directions of its potential applications.

## 2. Molecular Receptors Based on Porphyrin Conjugates with Macrocyclic Compounds of Different Natures

Interest in the chemistry of porphyrins was first sparked by the participation of these compounds in photosynthesis. However, it has now turned in another direction as porphyrins can be used as selective molecular receptors for certain types of substrates [15,16,17,18,19,20,21,22,23,24,25,26,27,28,29,30,31,32,33,34,35]. The fact that it is possible to chemically modify the tetrapyrrole molecule has enabled researchers to synthesize porphyrin conjugates with other macrocyclic compounds(porphyrin conjugates with calix[n]arenes [36,37,38,39,40,41,42,43,44,45,46,47,48,49,50,51,52,53,54], cyclodextrines [55,56,57,58,59,60,61,62,63,64,65,66], calix[n]pyrroles, fullerenes, ferrocenes, and resorcinarenes [66,67,68,69,70,71,72,73,74,75,76,77,78,79,80]) possessing their own properties, allowing them to form complexes with ions and organic molecules of different natures. Table 1 and Table 2 present some data on “molecular tweezer” macrocyclic receptors and the substrates they bind, based on results of the analysis of the most interesting works in this field.

Especially interesting among such compounds are covalently bound porphyrin-calix[4]arene macrocyclic receptors [8,10,36,37,38,39,40,41,42,43,44,45,46,47,48,49,50,51,52,53,54]. The first calix[4]arene-porphyrin conjugate capable of selectively identifying small organic molecules was described in [47].

By making *ortho*-carbomethoxyaldehyde (**1**) interact with pyrrole in the presence of trifluoroacetic acid, α-unsubstituted *meso*-aryl dipyrromethane (**2**) was synthesized (Scheme 1) which was thentransformedinto the respective 5,15-di-(carbomethoxyphenyl)-porphyrin (**3**) through condensation with benzaldehyde and subsequent oxidation of the intermediate porphyrinogen by 2,3-dichloro-5,6-dicyano-benzoquinone (DDQ). Hydrolysis of ester groups **3** subjected to the action of lithium hydroxide in tetrahydrofuran and subsequent interaction with oxalyl chloride in dichloromethane yielded porphyrin diacylchloride (**4**), which was later made to interact with the diamine of the respective calix[4]arene (**5**) in dichloromethane in the presence of catalytic amounts of triethylamine to produce the target porphyrin-calix[4]arene (**6**, M=H_2_). By boiling ligand **6** with zinc acetate in a methanol–chloroform mixture, the zinc complex of this compound (**7**, M=Zn) was obtained.

The ^1^H-NMR spectra of porphyrinate **7** in the presence of pyridine (Py) and imidazole (Im) have two rows of signals of aromatic protons belonging to the free and bound porphyrinate molecules of the substrate (Figure 1). The ring current shielding effect of the porphyrin macrocycle results in an up-field shift of the aromatic protons of the bound substrate molecules compared with the respective protons of the unbound substrate molecules.

The structure of the **7-Py** complex determined by the molecular mechanics methods using the MMFF force field of the MacroModel (V6.5) software package on the SGI O2 workstation is shown in Figure 2. Applying the method of spectrophotometric titration, it was established that porphyrinate **7** is most capable of forming complexes with imidazole (Table 3).

Thus, Zn-porphyrin **7** capped with calix[4]arene offers in its cavity a guest-binding environment for selective binding of small organic molecules, which makes it possible to distinguish even small structural differences between pyridine and 4-methylpyridine.Guests that are shaped in a way that is complementary to the confined cavity are capable of fitting into and binding to the cavity by van der Waals attractive interactions.

Another example of a calix[4]arene-capped porphyrin (**8**) was obtained by the authors of [48] on the basis of tetra-*tert*-butyl-tetrakis-(hydroxycarbonylmethoxy)-calix[4]arene (**9**) and tetra-(2-aminophenyl)-porphyrin (**10**) according to Scheme 2.

A zinc complex of compound (**11**) was obtained by treatment of ligand **8** with zinc acetate in DMF at room temperature. It was established that the modified porphyrin (**3**) shows very interesting ion-binding ability (Scheme 3). In NaClO_4_, Na^+^ is bound to the calix[4]arene cavity and ClO_4_^−^ exists as a counter anion to give the 429 nm band, whereas in KI, K^+^ is bound to the calix[4]arene cavity and I^−^ coordinates to Zn(II) to give 446 nm. These results indicate that the cavity with a hard-soft ditopic binding site is very suitable for recognition of I^−^ ions.

The synthesis and binding ability of a Zn-porphyrin (**12**) with a calix[5]arene cap were presented in [49]. In order to synthesize such a bridged porphyrin, 5,15-Bis-(carboxynaphtyl)-10,20-diphenylporphyrin (**13**) with syn arrangement of two naphthalene rings was prepared. Reaction of the porphyrin **13** with calix[5]arene with two amino functions (**14**) and subsequent treatment of the reaction product with zinc acetate gave the calix[5]arene-capped Zn-porphyrin **12** (Scheme 4).

It was shown that the calix[5]arene-capped Zn-porphyrin **12** is able to bind 4-methylpyridine much stronger that non-modified Zn-tetrphenylporphyrin due to the accommodation of the guest molecule into the calix[5]arene cavity. On the other hand, bulky substituents on the pyridine group, as in the case of 4-*tert*-butylpyridine, 4-phenylpyridine, and 3,5-dimethylpyridine, prevent the guest molecules’ penetration into the calix[5]arene cavity, and the constants of their binding to the modified receptor are commensurate with the corresponding binding constant to the Zn-tetraphenylporphyrin.

For the purpose of developing a novel photoelectronic active porphyrin derivative with good solubility and suitable selective recognition ability for the guest neurotransmitter dopamine (DA) molecule, a functionalized porphyrin **15** was designed (Scheme 5) by the reaction between 5-iodinephenyl-10,15,20-tris-(1-pyrene)-porphyrin **16** and propinyl-calix[4]arene in mixed trimethylamine/dichloromethane(1:1) solvent in the presence of PdCl_2_(PPh_3_)_2_ [10].

It was shown that a combination of the effective molecular recognition/enrichment and good optoelectronic properties derived from calix[4]arene and pyrene substituents as well as the *p*-conjugated porphyrin skeleton leads to the fluorometric and voltametric dual-mode sensing activity of the calix[4]arene-porphyrin **8** for DA in the range of 1–50 mM and 2–100 mM, respectively, with remarkable specificity, representing the first multi-modal molecular sensor for DA.

“Molecular tweezers” based on the thiacalix[4]arene-porphyrin conjugate **17** capable of selectively identifying the C_70_fullerene were obtained by the authors of [36].

The procedure of synthesis of the tetra-formyl derivative of thiacalix[4]arene in the 1,3-alternate conformation is presented in Scheme 6. Bromation of commercially available tetrahydroxy-thiacalix[4]arene (**18**) by *N*-bromosuccinimide in acetone was used to synthesize tetrabromo-derivative (**1****9**), which was then turned into the respective propoxy-derivative (**20**) through its interaction with propyl iodide in the presence of potassium carbonate in acetone. It should be said that introduction of bulky propoxy-substituents into the upper ring of the initial thiacalix[4]arene makes the molecule change its cone conformation to the 1,3-alternate one. The interaction of (**20**) with *tert*-butyllithium in tetrahydrofuran accompanied by cooling allowed preparation of the target thiacalix[4]arene tetraaldehyde (**21**), which was then condensed with monoamino-porphyrin (**22**) in dichloromethane in the presence of molecular sieves, to synthesize thiacalix[4]arene-Bis-(porphyrin) (**17**) (Scheme 7).

The ability of **17** to act as a receptor of the C_60_ and C_70_ fullerenes was studied by the authors using the ^1^H-NMR titration method. It was found that the C_70_ fullerene presence makes the signals of the porphyrin *NH*-protons shift up-field on the ^1^H-NMR spectrum. As the titration curve shows (Figure 3), the interaction leads to the formation of a 1:1 complex. It is shown (Table 4) that receptor **13** is highly selective to the C_70_ fullerene.

A calix[4]arene scaffolding was used by the authors of [38] to construct Bis-porphyrin (“jaws” porphyrin) hosts **23**–**28** (Figure 4) for supramolecular binding of fullerene guests. Fullerene affinities were optimized by varying the nature of the covalent linkage of the porphyrins to the calix[4]arenes.

Binding constants for C_60_ and C_70_ in toluene were explored as a function of substituents at the periphery of the porphyrin, and 3,5-di-tert-butylphenyl groups gave rise to the highest fullerene affinities (26.000 M^−1^ for C6_0_). The molecular modeling structure of **26-C_60_** is depicted in Figure 5. The origin of this high fullerene affinity has been traced to differential solvation effects rather than to electronic effects. Studies of binding constants as a function of solvent (toluene < benzonitrile < dichloromethane, cyclohexane) correlate inversely with fullerene solubility, indicating that desolvation of the fullerene is a major factor determining the magnitude of binding constants. A direct relationship between supramolecular binding of a fullerene guest to a Bis-porphyrin host and the appearance of a broad NIR absorption band has been established. The energy of this band moves in a predictable manner as a function of the electronic structure of the porphyrin, thereby establishing its origin in porphyrin-to-fullerene charge transfer.

The synthetic approach leading to the Pacman Bis-porphyrins was described by the authors of [51]. Established calixarene chemistry led to the diodideintermediate **29**, which was isolated exclusively in cone conformation. Sonogashira coupling of **29** and the Ni-porphyrin ethynyl derivative (**30**) gave corresponding mono- (**31**) and Bis-porphyrin-calixarene (**32**) conjugates. It should be noted that both **3** and **4** display a cone conformation in solution as observed by ^1^H-NMR. Further alkylation of **32** took place at a rather slow rate to yield the conformationally frozen cone derivative **33** (Scheme 8).

The same authors developed a versatile stepwise synthetic approach allowing selection of the calix[4]arene conformation, the type of functionalization of the calixarene rims, and the anchoring point of the chromophores on the calix[4]arene spacer [39]. Connection of the ethynyl-derivative **36** with the calix[4]arene derivatives **34**, **35**, and **38** in the presence of 10% Pd(PPh_3_)_2_Cl_2_, CuI, and Net_3_ in toluene gave three calix[4]arene Bis-porphyrin conjugates **37**, **39**, and **42** (Scheme 9).

It was shown that receptors **37**, **39**, and **42** display very different behaviors in the presence of bidentate substrate DABCO, with **37** allowing a fast exchange between bound and free substrates, while **39** and **42** provide effective bounding of DABCO.

The works [40,41,53,54] are also devoted to the synthesis and analysis of the complexing properties of Bis-porphyrin-calixa[4]renes. Bis-porphyrin-calix[4]arene conjugate (**43**) (Figure 6) was synthesized similarly to Scheme 9 from Bis-iodo-calix[4]arene and nickel *meso*-ethynylporphyrinate by the Sonogashira method [53].

Works [40,41,54] describe the synthesis of calix[4]arene-Bis-(porphyrin) **44** (Figure 6) through condensation of calix[4]arene-Bis-(5,5′-unsubstituted dipyrromethane) (**46**) with 5,5′-diformyl-dipyrromethane in dichloromethane in the presence of catalytic amounts of trichloroacetic acid (Scheme 10), with subsequent oxidation of the intermediate porphyrinogen with dichlorodicyanobenzoquinone. By heating of porphyrin ligand (**44**) and zinc acetate in dimethylformamide, a zinc complex of receptor (**45**) (Figure 6) was obtained.

The synthesis of calix[4]arene-Bis-(porphyrin) **45** through condensation of diformylcalix[4]arene (**48**) with 4,4′-dimethyl-3,3′-diethyldipyrrolylmethane and benzaldehyde in dichloromethane in the presence of catalytic amounts of trichloroacetic acid is depicted in Scheme 11. By modifying the lower ring of calix[4]arene platform **45** and subjecting it to the action of pentaethylene glycol Bis-tosylate in the presence of cesium carbonate, we obtained a macroheterocycle (**49**) containing three functional components: porphyrin, calix[4]arene, and a crown ether fragment (Scheme 12a). By modifying the lower ring of the calix[4]arene platform **49** we obtained a macroheterocycle (**50**) containing two ethoxycarbonyl groups in the calix[4]arene platform (Scheme 12b).

By applying the ^1^H-NMR method, we found that formation of complexes of calix[4]arene-Bis-(porphyrin) receptors **45**, **49**, and **50** with bidentate triethylenediamine (DABCO) represents the 1:2 type of interaction. Scheme 13 illustrates the interaction of receptor **50** with DABCO as an example. This hypothesis is also confirmed by the separation of the signals of the nonequivalent DABCO protons in the ^1^H-NMR spectrum of the complex (Figure 7a). It means that the mismatch between the geometry of the interporphyrin complexing cavity of the receptor and the substrate leads to one-center binding of the studied bidentate ligand and formation of an external complex (**51**) with the stability constant *K_a_* = 4.7 10^5^ M^−2^ [49].

However, in the presence of a potassium cation, the formation of a complex of receptor **52** with DABCO represents a 1:1 interaction (Scheme 14). This hypothesis is also confirmed by the clear singlet of the equivalent protons of the substrate in the complex ^1^H-NMR spectrum (Figure 7b). It means that the conformational changes caused by the complex formation of the macrocycle crown ether fragment with the potassium cation make the porphyrin fragments of the Bis-porphyrin receptor get closer to each other and produce a two-center 1:1 complex between **52** and the organic substrate and a stable internal complex (**53**) with the stability constant *K_a_* = 6.9 × 10^6^ M^−1^.

Several different approaches for the synthesis of Bis-porphyrin-calixarenes have been developed [37,42]. First, Bis-porphyrin-calixarenes (**54**–**55**), in which porphyrin moieties are linked to the calix[4]arene fragment through amide bridges, were synthesized by two methods: (i) by reaction of calix[4]arenes acyl chloride (**56**) with monoaminotetraphenylporphyrinininin the presence of base (Et_3_N); (ii) by a direct reaction of carboxylic acid (**57**) with monoaminotetraphenylporphyrin using dicyclohexylcarbodiimide as a coupling agent (Scheme 15) [42].

Later, Káset al. obtainedBis-porphyrin-calixarenescontaining directly linked porphyrin and calixarene fragments via condensation of an excess of pyrrole and *p*-methoxylbenzaldehyd with calix[4]arene-5,17-dialdehyde under BF_3_·Et_2_O catalysis in chloroform [37]. Subsequent oxidation of the intermediate porphyrinogen gave the corresponding Bis-porphyrin-calixarene conjugate (**60**) according to Scheme 16.

The ^1^H-NMR revealed the pronounced selectivity of the Bis-porphyrin derivative (**60**) towards C_70_ fullerene. Binding constants of the mono- (**58**) and Bis-porphyrin-calixarene (**60**) conjugates towards fullerene C_70_ are depicted in Table 5.

Calix[4]pyrrole was used in work [67] as the spacer holding the porphyrinate fragments in the face-to-face orientation to each other.

Thereaction of calix[4]pyrrole Bis-iododerivative (**61**) with zinc *meso*-ethynylporphyrinate (**62**) by the Sonogashira method was used to obtain calix[4]pyrrole-Bis-(porphyrin) conjugate (**63**) (Scheme 17). Applying spectrophotometric titration (Figure 8a), we established that formation of a complex of the calix[4]pyrrole fragment of receptor **61** with fluorine anions (Scheme 18) makes the Soret band in the absorption spectrum broader and less intensive, and in the region of the 1:1 molar ratio of the reagents, the reaction system becomes saturated (Figure 8b). It means that the interaction between receptor **63** and the anion produces 1:1 complexes (**64**) with the stability constant *K_a_* = 15,000 M^−1^.

The down-field shift of the *NH*-protons of the calix[4]pyrrole fragment indicates that the complex formation causes the spacer to change its 1,3-alternate conformation to a cone (Figure 9). It is shown that complex formation involving a calix[4]pyrrole fragment can be used to control complex formation through the interporphyrin complexing cavity of receptor **63**. Without the anion, the reaction of **63** with DABCO results in the formation of an external 1:2 complex (**65**) (Scheme 19) with the stability constant K_a_ = 4.7 × 10^5^ M^−2^. The formation of external 1:2 complex **30** is confirmed by the splitting of the substrate proton signals in the ^1^H-NMR spectrum of the respective complex.

Some of the DABCO protons located closer to the macrocycle experience the maximum shielding effect of its π-electron cloud, with their signals observed in the strong field region, whereas the signals of the other protons located further from the macrocycle appear in the weak field region [67].

However, it is shown that in the presence of the anion, the complex formation is accompanied by the formation of an internal 1:1 complex (**66**) (Scheme 20) with the stability constant *K_a_* = 6.8 × 10^6^ M^−1^. The formation of the internal complexes is confirmed by the fact that the spectrum shows only one signal of the substrate protons located in the strong field region. The same shielding of the substrate protons by the receptor porphyrin fragments leads to the appearance of signals of the equivalent protons experiencing the shielding effect of the porphyrin fragment π-electron cloud in the strong field region. Thus, the conformational changes caused by the formation of the calix[4]pyrrole fragment complex with the anion make the receptor porphyrin fragments move closer to each other and facilitate the formation of internal receptor–substrate complexes with two points of binding.

A supramolecular electron donor–acceptor ensemble composed of three different components was created on the basis of a porphyrin anion, radical cation of tetrathiafulvalene calix[4]pyrrole (67), and radical anion of Li^+^-encapsulated C_60_ (Li^+^-C_60_) (Figure 10) [9].

These components bind to one another in a specific fashion via a combination of electrostatic and donor–acceptor interactions (Figure 11). It was shown that binding of a porphyrin carboxylate anion to tetrathiafulvalenecalix[4]pyrrole results in electron transfer from **67** to Li^+^-C_60_ to produce the charge-separated state (**68**) in benzonitrile.

Upon photoexcitation of (**66**), photoinduced electron transfer from the triplet excited state of porphyrin carboxylate to **67** occurs to produce the higher energy charge-separated state (**68**), which decays to the ground state with a lifetime of 4.8 µs, as determined from the absorption recovery of the bleaching band at 650 nm ascribed to the porphyrin ground state (Figure 12). This study provides a new approach to creating noncovalent, multi-component systems that support the formation of photoinduced charge-separated states.

In work [44], the authors synthesized oxacalix-[2]-arene-[2]-Bis-(Zn-porphyrin) (**69**) through nucleophilic substitution of the AB_3_-Zn-porphyrin hydroxy-derivative (**70**) for 5,17-Bis-(methylsulfonyl)-oxacalixarene (**71**) (Scheme 21).

The ^1^H-NMR titration method showed that receptor **69** effectively binds the C_70_ fullerene molecules in a solution [44]. At the same time, the authors note that this receptor does not interact with the C_60_ fullerene. It means that if the C_70_ fullerene molecule fits into the interporphyrin complexing cavity of **69**, it creates the ideal conditions for substrate–receptor interactions leading to the formation of s Table 1:1 (**69**-**C_70_**) complexes (Figure 13) with the stability constant *K_a_* = 30,000 M^−1^.

The authors of [77] synthesized a stable supramolecular capsule (**72**) based on tetracarboxy-resorcinarene (**73**) and 2-pyridylporphyrin (**74**) (Scheme 22). The ^1^H-NMR method showed that receptor **72** can effectively bind small organic molecules of various sizes. Figure 14A shows the spectra of the porphyrin (a), resorcinarene (b), and supramolecular capsule on their basis (c). The spectrum (Figure 14A(c)) clearly shows that the formation of supramolecular complex **72** shifts the signal of the peripheral resorcinarene protons up-field. The fact that the biggest shift of the signals of the protons (H_i_) is observed in the resorcinarene -OCH_2_O- fragments located closer to the porphyrin macrocycle indicates that the resorcinarene fragment is located above the tetrapyrrole macrocycle.

Figure 14A(d) shows the ^1^H-NMR spectrum of the capsule **72** complex with an ethylene molecule. The strong field signal (in the region of −0.80 ppm) appearing in the spectrum when gaseous ethylene is purged through the receptor solution indicates the formation of an internal ethylene complex with the porphyrin capsule. It is shown (Figure 14B) that capsule **72** can selectively bind methane, ethane, propane, and ethylene molecules. The complex stability value (M^−1^, at 298 K) changes depending on the substrate molecule’s nature as follows: methane (56) < acetone (202) < ethylene (690) > ethane (527) > propane (236) > 2-methylpropane (3).

## 3. Molecular Receptors Based on Porphyrins Modified at the Macrocycle Periphery with Bulky Substituents

An important direction in the design of tetrapyrrole macrocyclic receptors for a certain substrate type is modification of the macrocycle periphery with bulky substituents or molecular fragments of different natures [81,82,83,84,85,86,87,88,89,90,91,92]. Bulky highly-branched lateral substituents are capable of forming additional complexing cavities that can be used for identification and selective binding of substrates of a certain type [93,94,95,96,97,98,99,100,101,102,103,104,105].

The authors of [103] obtained Zn-complexes of 5,15-diphenylporphyrins with alkoxy-substituents of various lengths in the macrocycle phenyl fragments (**75**–**77**) through condensation of dipyrromethane (**78**) with the respective aromatic aldehydes in the presence of zinc acetate and catalytic amounts of an acid, followed by oxidation of the intermediate porphyrinogens by *ortho*-chloranil (Scheme 23).

It is shown that porphyrinates form 1:1 complexes with imidazole (Im), propylamine (PrA), 1,3-diaminopropane (DAPr), and DABCO (Figure 15). The authors attribute the higher stability of complexes of dimethoxy-substituted porphyrinate **76** with propylamine (**76-PrA**, K_a_ = 25200 M^−1^) than that of the similar complex of diphenylporphyrinate **75** (**75-PrA**, K_a_ = 17800 M^−1^) to the presence of *ortho*-methoxy groups in its aryl fragments.

Despite the existence of two isomers of porphyrinate **76** with different positions of the alkoxy-aryl fragments relative to the tetrapyrrole macrocycle, both isomers can form receptor–substrate complexes with two binding sites (between the macrocycle zinc cation and the ligand nitrogen atom and between the macrocycle oxygen atom and the ligand hydrogen one), in contrast to diarylporphyrinate with a single-site interaction between the receptor molecule and the substrate (Figure 16).

The bigger increase in the stability constants of tetradodecyloxy-substituted porphyrinate **77** with DAPr (**77-DAPr**, K_a_ = 54200 M^−1^) than that in the respective complex of dimethoxy-substituted porphyrinate **76** (**76-DAPr**, K_a_ = 27400 M^−1^) is explained by the presence of alkoxy-substituents at both sides of the macrocycle coordination center (Figure 17). If the porphyrinate and substrate reaction centers match geometrically, complexes with three binding points are formed between the zinc dodecyl-substituted porphyrinate and Bis-amine [103]. The structure of the receptor–substrate complex was verified by the 2D ROESY NMR method.

It should be said that a lot of works describe the binding ability of macrocyclic receptors with well-defined geometries [105,106,107,108,109,110,111]. An analysis of the presented data indicates that their complexing ability is strongly dependent on the conformational mobility of the lateral substituents (molecular fragments) and the possibility of formation of intramolecular cavities and channels of various shapes within them.

The authors of [109] synthesized zinc porphyrinate (**79**) with first-generation carbazole branches by the reaction of azide-alkyne cycloaddition of *meso*-ethynylphenyl substituted carbazole (**80**) with the octa-azido derivative of zinc porphyrinate (**81**) in the presence of copper chloride (Scheme 24). In a similar way, they synthesized zinc porphyrinate (**82**) with peripheral dendrimer second-generation branching based on **81** and *meso*-ethynylphenyl substituted second-generation carbazole (**83**). Structures of the carazolylphenyl branches of the first- (**79**,**80**) and second- (**82**,**83**) generations are depicted in the Scheme 25.

It was established by the spectrophotometric titration and ^1^H-NMR methods that the formation of a complex between zinc porphyrinate (**79**) and triazole (TrA) (Scheme 26) leads to the formation of one family of spectral curves corresponding to its own set of isobestic points (Figure 18a).

The titration curve has one stage, which indicates the formation of one type of complexes of 1:1 composition (Figure 18b). The same results are obtained when TrA forms a complex with zinc porphyrinate with two and four carbazole branches (**84**,**85**) prepared in a similar way to **79** on the basis of di- and tetra-azido derivatives of the respective zinc porphyrinates.

In the case of di- and tetra-substituted porphyrinates (**84**,**85**) (Figure 19), the triazole-carbazole fragments are bound with the macrocycle by a benzene bridge, i.e., there are no bulky substituents in the *ortho*-position of the benzene bridge. Unsubstituted phenyl fragments in *meso*-tetraphenylporphin are known to possess sufficient conformational mobility and can rotate along the carbon–carbon bond linking the phenyl fragment with the tetrapyrrole macrocycle.

The stability constants of the respective complexes for zinc tetraphenylporphyrinate (**86**) and zinc porphyrinates with two (**84**), four (**85**) and eight (**79**) first-generation carbazole branches and the studied ligands are given in Table 6 and are shown in Figure 20 as a plot [109].

The data show that the most stable complexes are those formed with the porphyrinate with eight carbazole fragments (**79**). This is probably the result of the specific structure of the considered macrocyclic receptors.

Probably, in the case of di- and tetra-substituted structures, most of the carbazole branches are not fixed relative to each other but can quite freely rotate relative to the tetrapyrrole macrocycle. As a result, there are no clearly defined and spatially fixed complexing cavities in these porphyrinates. However, in the case of the octa-substituted porphyrinate (**79**), the aromatic branches are mostly fixed spatially due to the *ortho*-methyl groups in the benzene bridge linking the porphyrin and carbazole fragments and, probably, forming complexing cavities above and under the tetrapyrrole macrocycle plane. Due to the fact that the substrate fits well into the intramolecular cavity of the porphyrin receptor and that there are additional hydrogen bonds and/or π–π interactions between the ligand and the triazole fragments of the porphyrinate, the internal complex formed as a result is characterized by increased stability (Scheme 27).

Interesting results were obtained by the authors of [109] when studying the ability of carbazolyl-substituted zinc porphyrinates to form complexes with triethylenediamine, a two-centered substrate. Di- and tetra-substituted porphyrinates **84** and **85** were found to interact with DABCO, and the process consisted of two stages. The complex formation spectrum has two families of spectral curves (Figure 21), each with its own family of isobestic points. The logarithmic dependence of the changes in the optical density of the system on ligand concentration also has two stages (Figure 22).

The fact that the ^1^H-NMR spectrum of the complex formed in the region of the low ligand concentration has one signal of the bound DABCO protons (−3.4 ppm) indicates that the complex formed has a 2:1 composition and its stability constant K_a_ = 4.1 × 10^9^ M^−2^. The sandwich structure of the complex obtained when the ligand is located between the same porphyrin macrocycles makes all the ligand protons equivalent and leads to their appearance in the form of one signal in the strong field region.

The splitting of the proton signals in the complex ^1^H-NMR spectrum at higher ligand concentrations, the second stage on the titration curve (Figure 22), indicates that the complex formed has a 1:1 composition with the stability constant K_a_ = 6.9 × 10^6^ M^−1^. Some of the protons located near the macrocycle experience the maximum shielding effect of its π-electron system, and their signals are observed in the strong field region (−3.4 ppm). The shielding effect on the other protons located at a distance from the macrocycle is weaker, and their signals are observed in the weak field region (0.5 ppm). The porphyrinate with four branches interacts with DABCO in the same way but the complex formation occurs at higher ligand concentration. At the same time, it is shown that the complex formation of octa-substituted porphyrinate (**79**) with DABCO consists of only one stage. Evidently, the bulky spatially fixed substituents prevent two-point interaction with DABCO leading to the formation of 1:1 complexes only. The stability values of the **36-DABCO** and **39-DABCO** complexes are comparable.

The authors of [110] investigated the complex formation between carbazolyl-substituted zinc porphyrinates with first- (**79**) and second-generation (**82**) lateral branches and cryptands that are significantly different in their geometric and complexing parameters and the effect of alkali metal cations on this process. Cryptands are well known to be able to act as receptors of alkali metal cations. The binding selectivity is the result of the oxygen-containing three-dimensional complexing cavities in them allowing them to identify cations of different natures. Table 7 presents the binding constants of alkali metal cations by cryptand-[2,2,2] in methanol at 25 °C.

It should be said that complex formation is accompanied by considerable changes in the cryptand conformation, according to [111]. Figure 23 shows an example of complex formation between cryptand-[2,2,2] and a potassium cation and conformational changes accompanying this process in the complex.

On the other hand, cryptands themselves can act as substrates, for example, in the process of complex formation with metal porphyrinates. The cryptand nitrogen atoms and the porphyrinate metal cation act in this case as the binding centers. The authors of [110] studied the formation of a complex between zinc porphyrinate **82** and cryptand-2,2,2 (Cr1) as a model system. It is shown that the electronic absorption spectra of the reaction system have one family of spectral curves with its own set of isobesticpoints (Figure 24a). The titration curve has one stage showing that the complex formation process consists of one stage only (Figure 24b).

The complex stability constant value is shown in Table 8 and is generally lower than those of the zinc tetraphenylporphyrinate (**82**) complexes with nitrogen-containing bases.

As it has been mentioned above, the bulky carbazole substituents at the periphery of the tetrapyrrole macrocycle in octa-substituted zinc porphyrinate **79** lead to the formation of complexing cavities. A study of complex formation between the octa-substituted zinc porphyrinate with the cryptand showed that the stability of the 1:1 complexes (**79-Cr1**) that are formed as a result (Scheme 28) is much higher than that of the similar zinc tetraphenylporphyrinate complex with the cryptand (**82-Cr1**) (Table 6). Evidently, the intramolecular complexing cavity formed by the carbazole branches has a positive effect on the complex formation due to the good agreement between the substrate and receptor geometry.

The ^1^H-NMR data also indicate the formation of a 1:1 complex in the region of equivalent reagent concentrations (Figure 25). The spectrum of the initial ligand complex with the potassium cation has three signals of the substrate protons—the signals of the central (-OCH_2_CH_2_O-) and lateral (-OCH_2_- and -CH_2_N-) protons.

In the **79-Cr1** porphyrinate complex, the -OMH_2_- and -CH_2_N- proton signals split. The signals of some of the protons located closer to the macrocycle are observed in the strong field region. The others remain in the weak field.

The same authors compared the ability of first- (**79**) and second-generation (**82**) dendrimers to form complexes with cryptands with considerable differences in geometry (Cr1-Cr3, Figure 26), including the effect of the alkali metal cation (Li^+^, Na^+^, and K^+^) on the substrate–receptor interactions.

Figure 26 shows that the stability of the complex that has no peripheral zinc porphyrinate **86** substituents with Cr1-Cr3 cryptands is much lower than that of the respective complexes of dendrimers **79** and **82**.

In this case, dendrimer **82** with second-generation branches effectively binds the Cr1 and Cr2 ligands but does not interact with the Cr3 one, which is the largest in size. Cr3 is, evidently, unable to penetrate into the complexing cavity of the second-generation dendrimer because of its size.

A spectrophotometric study of the interaction between carbazolyl-substituted zinc porphyrinate **79** and Cr1 in the presence of alkali metal cations was conducted in a binary toluene–methanol solvent (9:1). The methanol additions are used to increase the solubility of lithium, sodium, and potassium bromides. Moreover, it should be taken into account that the alcohol additions weaken the coordination interaction between the porphyrinate zinc cation and the cryptand nitrogen atom. As a result, the stability of the receptor–substrate complexes formed becomes lower. It has been established that potassium bromide additions to the **79-Cr1** complex solution destroy the guest–host interaction (Scheme 29).

This hypothesis is, first of all, confirmed by the blue shift of the Soret band on the electronic absorption spectrum of the porphyrinate–cryptand reaction system (Figure 27a), indicating that the porphyrinate zinc cation does not interact with the cryptand nitrogen atom.

Secondly, the fact that the -OMH_2_- and -NCH_2_- proton signals in the NMR spectrum of the reaction system are not split (Figure 27b) also confirms the porphyrinate–cryptand complex dissociation in the presence of the alkali metal cation.

As Figure 28a shows, the presence of the lithium and sodium cations has practically no effect on the complex formation process considered. At the same time, the destruction of the **79-Cr2** complexes can only occur in the presence of lithium cations (Figure 28b).

Thus, complex formation inside the complexing cavity of the cryptand affects the formation of a complex between the carbazolyl-substituted zinc porphyrinate and the cryptand. The obtained experimental data can be explained by the effect of two factors. Firstly, as the calculations confirm, the formation of a complex between a cryptand and alkali metal cations is accompanied by changes in the molecule conformation as a whole. Secondly, the cryptands in the complexes with alkali metal cations are in the *endo-endo* conformation, in which the lone electron pairs of the nitrogen atoms face the ligand complexing cavity and, as a result, are inaccessible for coordination interaction with the porphyrinate zinc cation. The total effect of these factors, on the one hand, leads to the destruction of the porphyrinate–cryptand complex following the addition of a metal cation to the solution and, vice versa, prevents the zinc porphyrinate interaction with the cryptand of the corresponding metal cation.

It should be also said that the **79-Cr1** complex is destroyed as well when trifluoroacetic acid is added to the solution (Scheme 30). If the porphyrinate–ligand complex destruction under the action of the metal cation is caused by changes in the cryptate geometry in comparison with that of the cryptand, the complex destruction under the action of the acid is, evidently, caused by the breakage of the zinc–nitrogen bond between the porphyrinate metal cation and cryptand nitrogen upon protonation of the cryptand nitrogen atom.

The authors of [112] synthesized two-layered dendrimers (**87G_1_** and **87G_2_**) with inner carbazole and outer phenylazomethine layers by carrying out a reaction of zinc tetra(4-aminophenyl)porphyrinate (**87**) with the respective dendrons (**G1** and **G2**) in chlorobenzene in the presence of TiCl_4_ (Scheme 31).

The **87G_1_** and **87G_2_** receptors have binding centers of two types: (1) a complexing cavity formed by carbazole fragments of the dendrimer outer layer and (2) a zinc cation of the tetrapyrrole macrocycle. It was established that **87G_1_** and **87G_2_** exhibited selective complexing ability towards the **C_60_**, **C_70_**, and **C_84_** fullerenes. The binding represents a 1:1 type of interaction with the formation of complexes, in which one fullerene molecule is located in the complexing cavity made up of the carbazole fragments of the receptor outer layer. The stability constants of the respective complexes are given in Table 9. It has been established that in the case of **87G_2_**, the binding capacity of **C_60_** depends on the presence of phenyl-pyridine (PhPy) in the reaction mixture (the stability constant of the [**87G_2_-C_60_**]-**PhPy** complex is 1.5 times higher than the respective value of the [**87G_2_-C_60_**] complex).

According to [113], the formation of a complex between the porphyrinate zinc and the phenyl-pyridine nitrogen atom (Figure 29) causing conformational changes in the macrocycle dendrimer environment affects the ability of the outer carbazole layer of the **87G_2_** receptor to form complexes with C_60_.

## 4. Conclusions

The presented results show that porphyrin-containing macrocyclic compounds with a predetermined architecture and functional properties can be used to design ion-controlled molecular devices with a broad spectrum of action. The selectivity and high sensitivity of porphyrins to low-energy external effects make it possible to control chemical processes involving them. These characteristics together with the easily tunable shapes of molecules make the considered porphyrin-based molecular assemblies ideal receptors for extraction, directional transport, and sustained release of substances. Due to their extremely high extinction coefficients, porphyrins are also excellent molecular detectors for identification of processes affecting the tetrapyrrole macrocycle itself or the architecture surrounding it.

## Data Availability

Not applicable.

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
