# Peer review of "Macrocyclic Receptors for Identification and Selective Binding of Substrates of Different Nature"

_molecules, 2021, doi:10.3390/molecules26175292_

Round 1

Reviewer 1 Report

This review article entitled “Macrocyclic receptors for identification and selective binding of substrates of different nature” summarizes recent reports on the preparation and binding ability studies of porphyrin derivatives modified with calixarenes, calixpyrroles, oxacalixarenes, resorcinarenes, and with bulky carbozolylphenyl substituted dendrons. However, this manuscript seems to be premature for publication at this time. In fact, this reviewer feel that the structure of this manuscript should be reconstructed because of the following reasons.

First, several important studies such as ref. 88 are not discussed in the main text. Not only the authors’ works but also the pioneering works of other groups should be addressed carefully. Also, a summary table should be inserted to categorize a various types of the receptors met in the literature. As for ref. 47, the fact that the ring cavity distinguished the slight structural differences such as the pyridine over 4-methylpyridine should be emphasized. 

Second, the authors have to point out the novelty of the present review. Note that the concept of the cofacial bis-porphyrins with calixarene spacers was reported ~20 years ago (Org. Lett. 2002,4,2129). At this point, this reviewer can not find any conceptual or synthetic advances. 

Third, this manuscript lacks many of the important references to cite. Some of these are listed below.

Liu, Q. et al. New J. Chem. 2019, 43, 10376.

Yakushev, A. A. et al. J. Porphyrins Phthalocyanines 2019, 23, 1551. 

Holler, M. et al. J. Porphyrins Phthalocyanines 2011, 15, 1183.

Al-Azemi, T. F. et al. Tetrahedron 2011, 67, 2585.

Kas, M et al. Tetrahedron Lett. 2007, 48, 477. 

Arai, S. Chem. Lett. 2005, 78, 2007.

Dudic, M. et al. Tetrahedron 2003, 59, 2409. 

Dudic, M. et al. Org. Lett. 2003, 5, 149.

Iwamoto, H. Tetrahedron Lett. 2002, 43, 8191.

Dudic, M. et. Al. Tetrahedron 2002, 58, 5475.

Jokic, D. et al. Org. Lett. 2002, 4, 2129. 

Dudic, M. et al. Tetrahedron Lett. 1999, 40, 5949. 

Mibradt, R. et al. Tetrahedron Lett. 1995, 36, 2999.

Nagasaki, T. et al. Chem. Lett. 1994, 6, 989.

Asfari, Z. et al. Tetrahedron Lett. 1993, 34, 627.

Pognon, G. et al. J. Am. Chem. Soc. 2006, 128, 3488. 

In addition, many of the references are not properly cited. For instance, refs 6, 7, 8, 9, 10, 15, 21, 29, 30, 31, 32, 33, 57, and 68 do not correspond to the sentences. 

Finally, this reviewer suggests several minor points as follows; 

In Scheme 3 and Figure 3, the structure of Y should be clarified.

In Schemes 7 and 8, “tolyene” should be replaced by “toluene”.

The title of ref. 61 is missing.

There are typos in the captions of Figures 5, 12, 15, 18, 24, and Table 3.

In the caption of Figure 14, “ROSY” should be replaced by “ROESY”.

The chemical structures in Figure 23 are duplicately pasted.

This reviewer is wondering what the asterisks mean in Figure 24.

The description of the repeated substructures of the carbazole moieties is not correct. Please correct the places of the brackets.

The corresponding 1D NMR data should be pasted in the vertical and the horizontal lines in the 2D NMR spectra in Figure 14, otherwise the readers cannot follow what the cross-peaks mean.

The quality of the figures should be improved.

Reviewer 2 Report

Manuscript entitled “Macrocyclic receptors for identification and selective binding of 2 substrates of different nature”, by Galina Mamardashvili, Nugzar Mamardashvili and Oscar Koifman, discussed the results of studies in the field of synthesis and receptor properties of porphyrins modified with calix[4]arenes, calix[4]pyrroles, oxacalix-[2]-arenes-[2], resorcinarenes and with bulky carbazolylphenyl substituents of different generations capable to form inclusion complexes with substrates of different nature. This review is not very clear, and these references are very old. 1.It is very necessary to add some new references based on the synthesis and receptor properties of porphyrins. 2.Some H NMR spectra should be deleted, in addition, Scheme needs to be described. 3.In the abstract and introduction sections, the author needs further condense and summarize.

Reviewer 3 Report

The manuscript by Oscar Koifman reports the review of porphyrin-based receptor system using supramolecular interactions. A series of conjugates of porphyrins and macrocycles toward the binding of ions and molecules were summarized. Especially, calix arenes/pyrroles and so on are used for the receptors, while a carbazolylphenyl group as a different-type of functional molecule is employed. In the introduction, the authors briefly described these differences. However, the general and common concept should be described more and displayed by a figure, which will be helpful for the readers from other fields. Thus, this reviewer major modification before publication in Molecules. The following comments are minor points to be revised.

  1. A lot of mistakes of numbering the compounds in Schemes and Figures are found. It cause to be difficult to evaluate the correction of the manuscript. Please carefully check and correct them.

  1. Some descriptions in schemes are very bad. For example, structure were overlapped in Scheme 3; Only the Ni complex is described in Scheme 4 but M is used for Ni, H and Zn; Only freebase is described in Scheme 5 and so on.

  1. Several labels in the Figures are not good. For example, a is left side but b is right side in Figure 6; a is doubled in Figure 26 and so on.

Round 2

Reviewer 2 Report

The manuscript can be accepted in the present form.

Author Response

Thanks a lot for your positive comment.

Reviewer 3 Report

The manuscript is greatly improved. This reviewer agrees with publishing in Molecules.

Author Response

Thanks a lot for your positive comment.